# A toxin/antitoxin system targeting the replication sliding-clamp induces competence in *Streptococcus pneumoniae*

Mickaël Maziero[1,2], Dimitri Juillot[3], Isabelle Mortier-Barrière[1,2], Rut Carballido-Lopez[3], Nathalie Campo[1,2], Pierre Genevaux[1,2], Patricia Bordes[1,2], Patrice Polard[1,2]*, Mathieu Bergé[1,2]*

1 Laboratoire de Microbiologie et Génétique Moléculaires (LMGM), Centre de Biologie Intégrative (CBI), CNRS, Toulouse, France, 2 Université de Toulouse, Toulouse, France, 3 Université Paris-Saclay, INRAE, AgroParisTech, Micalis Institute, Jouy-en-Josas, France

* patrice.polard@utoulouse.fr (PP); mathieu.berge@utoulouse.fr (MB)

## Abstract

*Streptococcus pneumoniae* is a pathogenic bacterium capable of entering a cellular differentiation state, called competence, which enables it to acquire new genetic functions by natural transformation, as well as physiological functions such as tolerance to a number of antibiotics. The transition to this state is regulated by various environmental or intracellular signals that converge on the *comCDE* operon, which groups together the competence initiation genes. A fraction of activated cells is sufficient to propagate competence to the whole population via the product of the *comC* gene, the competence stimulating peptide (CSP). Remarkably, depletion of the essential ClpX/ ClpP AAA+ protease has been shown to induce the *comCDE* operon. Here we demonstrate that the ClpX-dependent induction of competence relies on the Spr1630 toxin (RipA), part of a Rosmer toxin-antitoxin system. We show that this toxin generates replicative stress by acting on the sliding clamp of replication, inducing transcription of the *comCDE* operon. Bacteria that produce RipA appear to lose their viability but remain metabolically active and able to produce CSP, thereby transferring competence to viable neighbouring cells.

## Author summary

The environment in which bacteria live puts them under a great deal of stress, forcing them to adapt constantly, either temporarily or permanently. *Streptococcus pneumoniae*, a pathogenic bacterium implicated in various pathologies such as otitis, meningitis and pneumonia, is also subject to stress, whether from its host, antibiotic treatments or the microbiota in which it lives. In response to this, *S. pneumoniae* is able to switch to a differentiated state called competence. This allows it to acquire new genetic characteristics through natural transformation, but also to better tolerate stresses such as antibiotics pressure.

**Data availability statement:** All relevant data are within the paper and its Supporting information files.

**Funding:** This work was funded by the Agence Nationale de la Recherche; grants ANR-17-CE13-0031 (to P.P.) and grants ANR-24-CE35-4598 (to P.B. and M.B). M.M. was supported by an Ministère de l'Enseignement Supérieur et de la Recherche phD fellowship. The funders had no role in study design, data collection and analysis, decision to publish, or preparation of the manuscript.

**Competing interests:** The authors have declared that no competing interests exist.

The underlying signals and signaling pathways of this phenotypic switch remain poorly characterized. In this study, we identified a novel toxin–antitoxin system that, when activated, causes a subset of the population to self-sacrifice by disrupting its own DNA replication. This self-induced arrest serves as a signal that promotes the transition to competence in neighboring cells, thereby improving the capacity for adaptation at the populational level.

## Introduction

*Streptococcus pneumoniae* is a commensal bacterium of the human nasopharynx, but can become an invasive or non-invasive pathogen leading to a wide range of diseases such as otitis, pneumonia, meningitis [1,2]. The emergence of antibiotic resistance in *S. pneumoniae* is making treatment more complex, and led to the deaths of around 600,000 people in 2019 [3]. This is one of the reasons why *S. pneumoniae* was ranked among the 12 priority pathogens by the World Health Organization in 2017. Several studies have strongly suggested that the emergence and rapid spread of antibiotic resistance is due to the ability of *S. pneumoniae* to undergo natural transformation [4].

Natural transformation is a horizontal gene transfer mechanism that enables bacteria to capture exogenous DNA and integrate it into their chromosomes through homologous recombination. First identified in *S. pneumoniae* [5,6], then in a wide range of bacteria, this mechanism has been relatively well characterized at the molecular level in a variety of bacteria [7–9]. In *S. pneumoniae*, most of the proteins required for natural transformation are only expressed during a period when the bacterium enters a particular physiological state called competence. The induction of competence is regulated by the level of transcription of two operons, *comCDE* and *comAB*. The *comC* gene encodes for a peptide [10] which is exported and matured by the ABC-transporter ComAB; the externalized product is called Competence Stimulating Peptide (CSP). CSP is able to activate the ComDE two component system [11]. In turn, phosphorylated ComE activates transcription of the *comAB* and *comCDE* operons as part of the early competence regulon, inducing an autocatalytic loop [11,12]. Two waves of genes are then successively induced: late and delayed competence genes [13,14], resulting in ~14% of the transcriptome being modified during competence.

The transition to competence is initially an individual behavior and when a sufficient fraction of the population has reached competence, the entire population is converted to competence following transmission of CSP via cell contact [15] by the self-inducing fraction [16]. The time required for this inductive fraction to appear is highly dependent on the stimuli received individually by the cells, which will affect the transcription of the early *comCDE* operon. Several stresses are known to promote competence, such as drugs that alter replication or genome integrity [17–19], temperature [20,21] and ribosome decoding [22]. Nonetheless, it remains evident that additional pathways have yet to be identified, and that some of the pathways

currently known are still insufficiently characterized at the molecular level. Early on in the molecular characterization of competence, it was noted that ClpP appeared to repress its development [23]. The *clpP* gene encodes the proteolytic subunit that forms the Clp ATP-dependent protease complex, together with an ATPase subunit ClpC, ClpE, ClpL, or ClpX. Systematic CRISPRi (clustered regularly interspaced short palindromic repeats interference) of *clpP* or any of the genes encoding the ATPase subunits demonstrated that only depletion of ClpP or ClpX induced competence development [24]. It was logically proposed that ClpX is the main ATPase implicated with ClpP to repress competence. Interestingly, ClpX is the only ATPase subunit of the ClpP complex that is essential for *S. pneumoniae* viability [25]. The essentiality of *clpX* was shown to be due to the presence of the gene *spr1630* which is proposed to encode the toxin of a so far uncharacterized toxin-antitoxin (TA) system [26].

The aim of the present work was to characterize the molecular pathway inducing competence under the control of ClpXP. Using genetic approaches, we demonstrated that competence induction by depletion of ClpX but not ClpP is dependent on *spr1630*. We demonstrated that Spr1630 is able to induce competence alone. We experimentally demonstrated that Spr1629 antagonizes the effects of Spr1630 leading us to propose that Spr1630 and Spr1629 form a toxin-antitoxin system (TA) that belongs to the large Rosmer TA family. Furthermore, our data strongly suggest that Spr1630 targets DnaN, the sliding clamp in DNA replication, and as such we propose to rename this TA system RipAB (Replication Interfering Protein). Finally, we place the impact of this TA system in the population context of the onset of competence and show that production of the RipA toxin gives rise to an inducing fraction that acts as a group of sentinels that warn the rest of the population of a stress by propagating competence development.

## Results

### Competence induction by *clpX* depletion is RipA-dependent

We first sought to reproduce the induction of competence by transcriptional depletion of *clpX* or *clpP* as described previously [23,24]. To monitor the development of competence, we used a transcriptional fusion of the *comCDE* early operon promoter with the luciferase gene as a reporter [17,27]. We then constructed strains carrying constructions that allow Clustered Regularly Interspaced Short Palindromic Repeats interference (CRISPRi) in *S. pneumoniae* [24]. Briefly, the defective *Streptococcus pyogenes* Cas9 (dCas9) controlled by an IPTG-dependent promoter was integrated into the *S. pneumoniae* chromosome together with the *lacI* gene and a single-guide RNA (sgRNA), targeting *clpX* or *clpP*, under the control of constitutive promoters [24]. When these cells were cultured in an unfavourable environment for the spontaneous development of competence (pH 7.0), no changes in the transcription of the *comCDE* operon were observed (Figs 1A and S1A, black lines). The addition of IPTG induced a strong increase in luciferase production reflecting the induction of transcription of the *comCDE* operon in either ClpX and ClpP depletion (Figs 1A and S1A, red lines), suggesting that the depletion of ClpX or ClpP is a competence trigger, as previously published [24]. Notably, the increase of *comCDE* transcription was concomitant with a reduction in growth, which reports on the depletion of the Clp proteins. Since the biological essentiality of ClpX has previously been linked to the *ripA* gene [26], we explored the possibility that RipA is also involved in triggering competence. As expected, deletion of *ripA* significantly improved growth of ClpX depleted cultures (compare Figs 1B–1A and S1B–S1A). Importantly, competence induction was completely abolished in the ClpX depleted culture lacking *ripA* (Fig 1B, *top left panel*), indicating that competence depends solely on RipA when ClpX is depleted. In an interesting way, and in contrast to the ClpX depletion background, ClpP depletion was still able to trigger competence in the *ripA* null mutant strain (compare Fig 1B, top right panel to Fig 1A, top right panel), suggesting that ClpP dependent induction of competence is not functionally related to RipA. However, as the absence of *ripA* slightly enhanced growth of the ClpP-depleted strain (S1C Fig), we cannot rule out the involvement of ClpP in RipA toxicity management. Altogether, these results confirmed the link between ClpX and RipA, and revealed a ClpX-dependent role of RipA in competence induction.

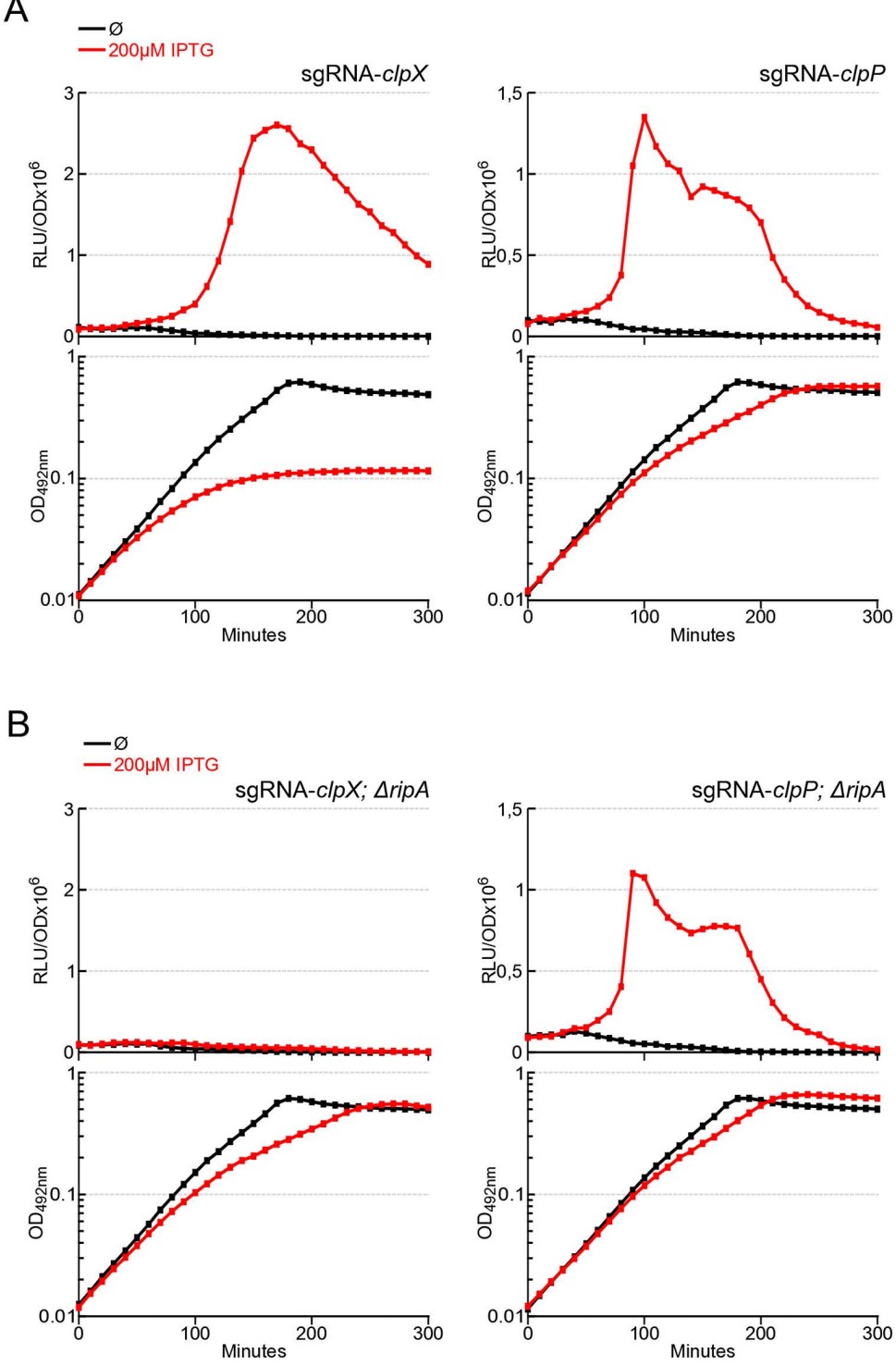

**Fig 1. Depletion of *clpX* or *clpP* induces competence.** A. *comCDE* expression was monitored in strains carrying CRISPRi guide RNA targeting *clpX* R4993 (left) or *clpP* R5203 (right). Cells were grown in C+Y medium at 37°C with or without IPTG, as shown in the colour key. Culture was initiated 100 minutes prior to the first measurement at time 0. *comCDE* expression values are expressed in relative light units per OD (RLU/OD) in the top panel and the corresponding growth curves are reported as $OD_{492nm}$ in the bottom panel. B. Identical to A in *ΔripA* strains carrying CRISPRi guide RNA targeting respectively 4995 (*clpX* depletion, left) and R5250 (*clpP*, right). For the sake of clarity, only a single data set, representative of at least three independent experiments carried out on different days, is presented. Compilation of these triplicate is presented in S1 Fig.

## The RipA toxin induces competence and is antagonized by the RipB antitoxin

To test the ability of RipA to induce competence, we produced the protein under the control of an IPTG inducible promoter [28]. To ensure tight repression of the IPTG-sensitive promoter, a second copy of the *lacI* gene driven by a constitutive promoter was integrated into the *S. pneumoniae* chromosome [24]. As *ripA* is part of an operon together with the essential *ripB* gene (*spr1629*) [29](Fig 2A), we also constructed strains containing the whole operon or *ripB* alone under IPTG induction. Figs 2B and S2 show that expression of *ripA* induces competence development in an IPTG-dependent manner whereas *ripB* expression does not. Interestingly, expression of the *ripAB* operon does not lead to competence development either, suggesting that *ripB* expression inhibits the action of RipA. To reinforce this observation, we assessed whether *ripA* or *ripAB* mutants exhibit altered competence-induction dynamics under our culture conditions. Cells were grown under mildly acidified conditions, in which competence becomes increasingly infrequent, and the proportion of cultures initiating competence was quantified. The wild-type strain displayed a gradual decline in competence entry. Notably, *ripA* and *ripAB* deletion mutants lost this capacity more rapidly than the reference strain S2B Fig. These findings are compatible with the notion that RipA may be a relay that allow integration of an additional signal that could enable the wild type to initiate competence more effectively than the mutants.

To better resolve the impact of RipA production on competence development, we assessed how increasing RipA levels affect the basal expression of the *comCDE* operon. For this purpose, we repeated the experiment in a *comC0* background to prevent activation of the autocatalytic loop. Basal *comCDE* expression increased proportionally with RipA production (Fig 2C). We further examined *comA*, another member of the same regulon, but its basal expression showed only a modest and nonsignificant response (S3A Fig). In contrast, the basal transcription level of tRNA-Arg5, which is not part of the competence regulon, also increased upon RipA production and mirrored the *comCDE* expression profile (S3A Fig). Because tRNA-Arg5 and *comCDE* are both located near the replication origin, these findings are consistent with competence induction driven by replication stress and the overexpression of origin-proximal genes [19].

Across all tested genetic backgrounds, we also note that *ripA* expression alone causes growth retardation in correlation with IPTG concentration and again co-expression of *ripB* cancels this growth defect (Fig 2B). We confirmed this by assessing the growth capacity of these strains on agarose medium in presence of increasing IPTG concentrations (Fig 3AB). From 10 µM IPTG, the strain expressing only *ripA* began to lose viability. Growth ability was almost completely lost from 20 µM IPTG. Production of *ripB* in operon with *ripA* completely cancelled out this toxicity. To test whether RipB could exert its protective effect in trans, we expressed *ripA* and *ripB* under IPTG inducible promoters, but at two different chromosomal *loci*. In this condition, RipB again antagonised the effect of RipA (Fig 3C). These results are consistent with the idea that RipA and RipB form a TA system as previously proposed [26]. We next tested whether RipA could be responsible for the previously observed essential nature of *ripB* [30] by attempting to delete *ripB* in different genetic backgrounds. Transformation results shown in Fig 3D demonstrate that *ripB* deletion could be obtained only when *ripA* was absent, further supporting that RipAB is a *bona fide* TA system in *S. pneumoniae*. Since RipA is responsible for the essentiality of both *clpX* [26] and *ripB* in *S. pneumoniae*, these data suggest that both proteins are controlling RipA activity. To explore this hypothesis, we tested whether RipB overexpression could compensate for the absence of ClpX. To do so, we transformed a strain capable of overexpressing RipB under the control of the LacI promoter with a *clpX* deletion. Under these conditions, only the strain plated in the presence of IPTG was able to grow (S3B Fig). This provides strong evidence for a functional link between ClpX and RipB (see discussion).

## Suppressor mutations in *dnaN* antagonize RipA toxicity

In search for RipA potential target(s), we took advantage of a *S. pneumoniae* interactome study based on yeast two-hybrid (Y2H) approach, which suggests that RipA interacts with DnaN [31]. As DnaN encodes the sliding clamp, an essential element in DNA replication, we considered that DnaN could be the target of RipA. To test this hypothesis, we searched for genetic suppressors of RipA toxicity in the *dnaN locus*. To this end, the strain producing RipA under the control of an

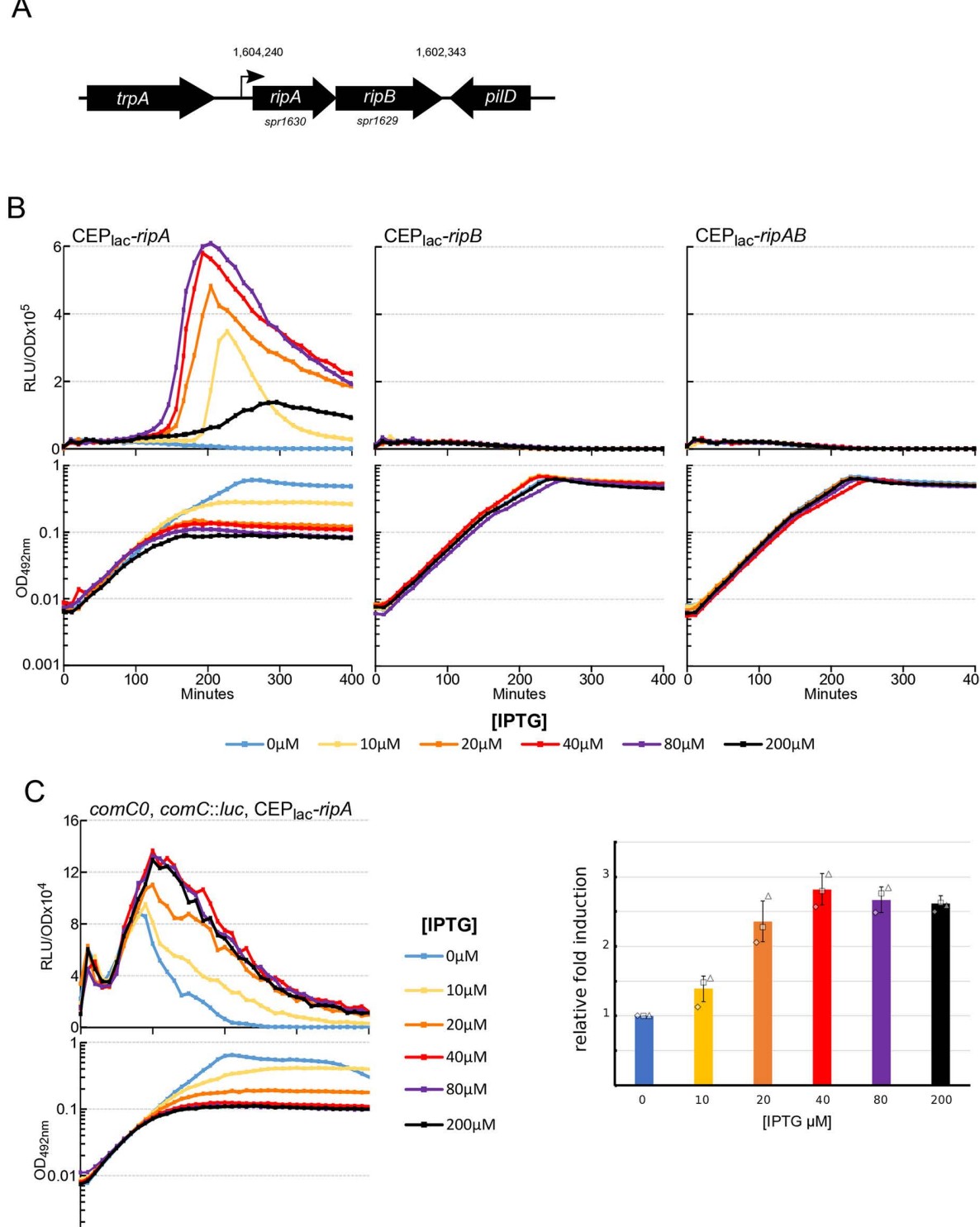

**Fig 2. *ripA* expression induces competence development.** A. Schematic representation of *ripA-ripB locus* and its chromosomal surroundings. Numbers indicate the first and last nucleotide positions of the *ripAB* operon in the *Streptococcus pneumoniae* R6 genome (GenBank: AE007317.1). B.

*comCDE* expression was monitored in strains expressing *ripA* (R5139), *ripB* (R5138) or *ripA-ripB* (R5140) under the control of an IPTG inducible promotor. Cells were grown in C+Y medium at 37°C with increasing concentration of IPTG from the first measurement at time 0. Top panels: luciferase activity expressed in relative light units per OD (RLU/OD). Bottom panels: corresponding growth curves. C. Induction of basal transcription levels of *comC*. *comC0*-derived strains producing RipA under an IPTG-inducible promoter (R5089) were grown in the presence of increasing IPTG concentrations. *comC* transcription levels were monitored through luciferase activity (left panels). For clarity, only one representative experiment is shown; however, the areas under the curves were quantified from three independent experiments and are presented as bar graphs with their respective standard deviations as shown in the middle panels. Right panels: corresponding growth curves.

IPTG-inducible promoter was transformed by error-prone~4kb PCR fragments centred on the *dnaN locus* and plated on agar medium supplemented with 20 µM IPTG. Interestingly, transformation with error-prone *dnaN* PCR fragments resulted in a higher number of colonies (about tenfold) compared to non-transformed control cells (Fig 4A). Sequencing the 4kb *dnaN* region of these transformants revealed that all mutations were located in the *dnaN* gene (Fig 4B). Seven strains with single mutations were obtained, leading to seven different amino acid substitutions affecting 5 different residues of DnaN (H183, L185, Y249, Y333, L371). Remarkably, these five residues are clustered in the same region of the DnaN structure (Fig 4B), a hydrophobic pocket implicated in DnaN protein-protein interactions [32]. We then characterized the ability of these mutations to suppress the toxicity linked to increased production of RipA (Figs 4C and S4). We found that, the majority of the mutations allow significant growth at IPTG concentrations similar to the chosen selection threshold (20 µM IPTG). Beyond this concentration, their effect diminishes drastically as *ripA* transcription increases. However, two mutations (Y249N) and (Y333H) enable growth at the highest IPTG concentration tested (200µM IPTG). This suggests that the various mutations do not exhibit the same ability to prevent toxicity of RipA. We have therefore categorised them as strong (Y249N, Y333H), intermediate (L185R) or weak (H183Q, H183L, H183P, L371F) suppressors.

## DnaN substitutions impair DnaN-RipA interaction in yeast two-hybrid assay

Since RipA interacts with DnaN in yeast two-hybrid assays (Y2H) [31], we investigated whether the single nucleotide polymorphism suppressive mutations in *dnaN* alter this interaction. We decided to test this interaction with one of the strongest suppressive mutations (Y249N), as well as a representative of the weaker mutations (H183P), and a mutation that displays an intermediate suppressive phenotype (L185R). First, we detected an interaction between DnaN with itself as expected, but also for RipA with itself, suggesting a potential dimerization of the protein RipA (Figs 5A and S5). We also reproduced the Y2H interaction between RipA and DnaN in both directions of interactions. However the RipA:DnaN interaction was completely abolished whatever the suppressive mutation analysed, (Figs 5B and S5) suggesting a loss or reduction of physical interaction between RipA and DnaN proteins that express suppressive mutations. Together, these data suggest that RipA targets DnaN through a physical interaction. To evaluate whether this interaction could also be detected directly in S. pneumoniae, we employed a split-luciferase assay. The small luciferase subunit was fused to the N-terminal region of DnaN, while the large subunit was fused to N-terminal region RipA, and both fusions were expressed from independent, inducible platforms. To quantify background activity, we measured the non-specific interaction signal in control strains expressing the small subunit fused to either RipA or DnaN, while the large subunit was produced independently. A detectable luminescence signal was observed when both fusion proteins were co-expressed (Fig 5C). In contrast, DnaN variants carrying the suppressor mutations displayed a markedly reduced signal (Fig 5C), further supporting an interaction between RipA and DnaN.

## DnaN substitutions impair competence development mediated by RipA

We next investigated whether the interaction between RipA and DnaN was responsible for the induction of competence in cells overexpressing RipA. To answer this, we tested the induction of competence in strains carrying the three categories of suppressive mutations described above. All strains carrying a suppressive mutation showed

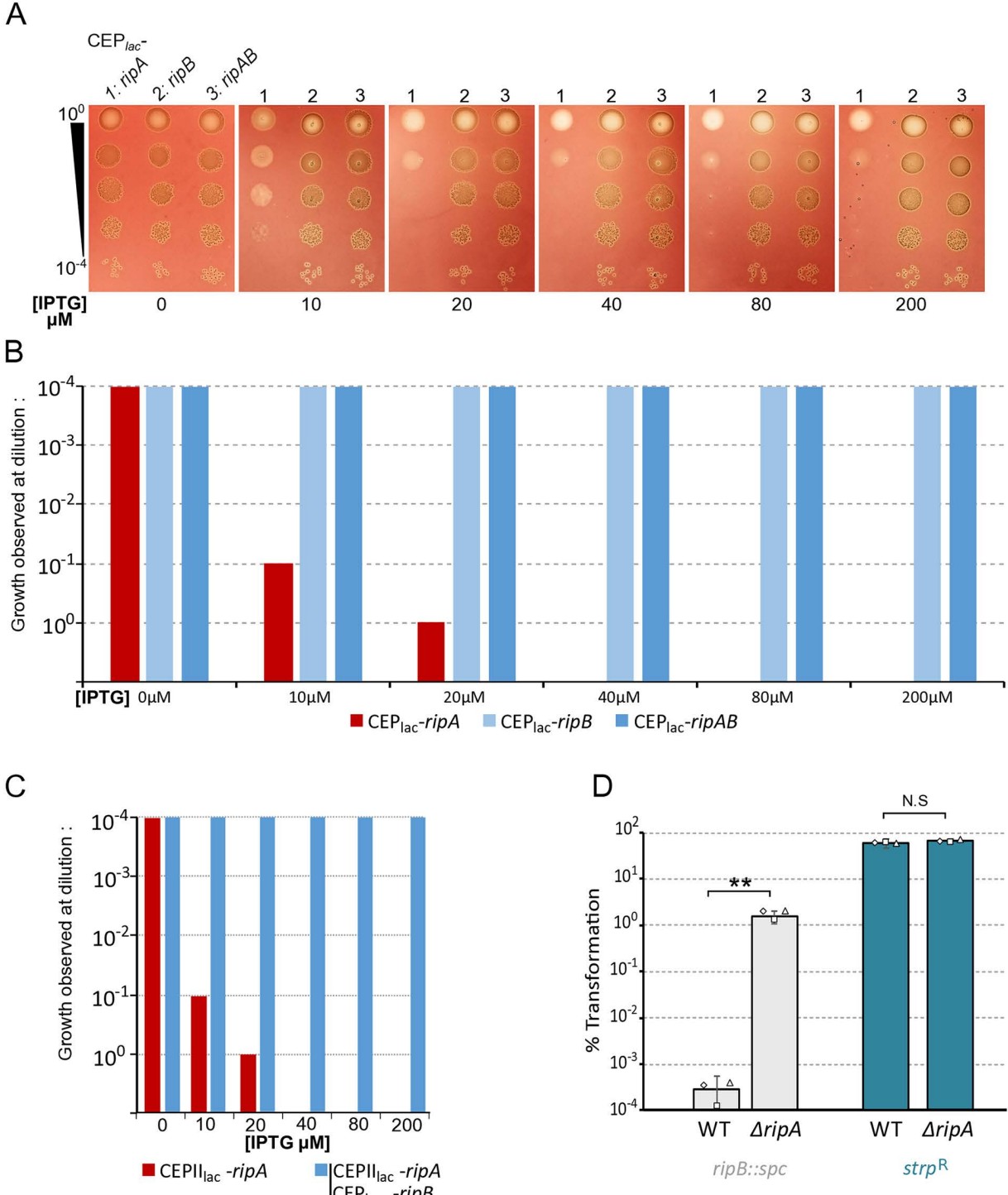

**Fig 3. Trans-dependant suppression of RipA toxicity by RipB.** A. Strains expressing *ripA* (R5139), *ripB* (R5138) or *ripA-ripB* (R5140) under the control of an IPTG inducible promotor were serially diluted and spotted on CAT agar supplemented with 4% horse blood containing different concentrations of IPTG. Plates were incubated at 37 °C overnight. For clarity, only one representative spot assay is shown. Three independent spot assays performed on different days produced similar results, which are quantified in panel B as a histogram to indicate plating efficiency (n = 3). C. Same as B. for strains expressing *ripA* under the control of CEPII_lac IPTG inducible promotor at the cps *locus* (see material and methods), (R5239 and R5259). In strain R5259, *ripB* is expressed under the control of CEP_lac IPTG inducible promotor at the "ami locus" (n = 3). D. Percentage of transformation of strain wild type (wt)

(R800) or invalidated for *ripA* (R4796) with DNA carrying *ripB* invalidation (spectinomycin insertion). In the control, these two strains were transformed with DNA carrying a point mutation conferring resistance to streptomycin (*rpsl41*). Diamonds, squares, and triangles indicate the dispersion of the experimental results.

better growth in liquid medium compared to the reference strain in IPTG+ conditions (Figs 6A and S6). Mutant strains displayed a gradation in the suppressive effect of the toxicity, as observed in spot test assays (Fig 4C). Thus, the strain carrying the most effective suppressive mutation (DnaN$^{Y249R}$) did not show any growth problems whatever the IPTG concentration. Conversely, the weakest suppressor mutant (DnaN$^{H183P}$) remained quite strongly impacted at the highest RipA production level. The DnaN$^{L185R}$ mutant exhibited an intermediate growth behaviour (Figs 6A and S6). Overall, competence induction of the mutant strains was strikingly impaired compared to the WT reference strain (Figs 6A and S6). However, enlarging the *comCDE* transcription observation scale by a logarithm (Fig 6B) revealed a gradation in behaviour during the development of competence in the three representative mutant strains. Indeed, DnaN$^{Y249R}$ never induced competence whatever the concentration of IPTG whereas, DnaN$^{H183P}$ and DnaN$^{L185R}$ strains only displayed a residual competence burst at high ITPG concentration (Fig 6B). It is interesting to note that the ability to trigger competence appears inversely proportional to the categorisation proposed to describe the ability to withstand the toxicity of RipA.

### "Doomed cells" are able to promote competence of naive cells

All the above elements support a link between the toxicity of RipA and its ability to induce competence. To test this, we compared the toxicity and competence induction kinetics. When RipA-producing cells were observed by microscopy, it was clear that bacterial multiplication was affected after 90 minutes of IPTG exposition, but that cell integrity was maintained for 30 additional minutes (Figs 7A and S7AB). To complete this observation, we evaluated the ability of the same culture to generate colonies on IPTG-free agarose medium after different RipA exposure duration. Fig 7B shows that most cells lost the ability to form colonies after 45 minutes of exposure. Yet, since competence develops around 120 minutes of RipA-induced toxicity (Fig 2B), we examined whether these cells remain functionally active, at least regarding their competence induction profile. To test this, RipA-producing cells were exposed to IPTG for over 60 minutes and then treated with synthetic CSP to rapidly trigger competence. These cells carrying the *comC*::*luc* reporter gene were still able to respond to CSP at a higher level than unstressed cells that did not produce RipA (Fig 7C upper panels). We hypothesized that this over-transcription could be due to a possible effect of RipA on DNA replication. To test this, we exposed cells to 6-(p-hydroxyphenylazo)-uracil (HPUra), a compound well known to induce replication fork stalling [33]. Very similar results were obtained (Fig 7C upper panels). When the same experiment was conducted under CSP-free conditions, reflecting the baseline *comCDE* levels, a comparable induction was again observed (Fig 7C lower panels). Taken together, these results support the notion that RipA influences replication. Finally, to demonstrate that RipA could interfere with replication, we monitored the behaviour of DnaX, the clamp loader, through a protein fused to the fluorophore (here YFP), as previously established [34,35]. At the early time point (t = 30 minutes), most cells exhibited a single mid-cell YFP–DnaX focus. After 180 minutes, fluorescence microscopy revealed a significant increase in cells lacking YFP–DnaX foci (bleu arrows) in cultures producing RipA (+IPTG), along with cells displaying mislocalized foci (green arrows) (S7C Fig). Quantification of these populations revealed a gradual increase in both cell categories (S7DE Fig), with cells exhibiting polar foci emerging earlier than those completely devoid of foci, however without allowing us to establish a direct link (S7 Fig). But, taken together, these observations further support the hypothesis that RipA interferes with the replication process.

We then tested if RipA-producing cells, were still able to produce enough CSP to propagate competence to naive cells, despite their condition. For this, we grew a strain expressing *ripA* under IPTG promoter but lacking the *comC::luc* reporter for 120 minutes in medium containing IPTG to allow induction of RipA-mediated competence.

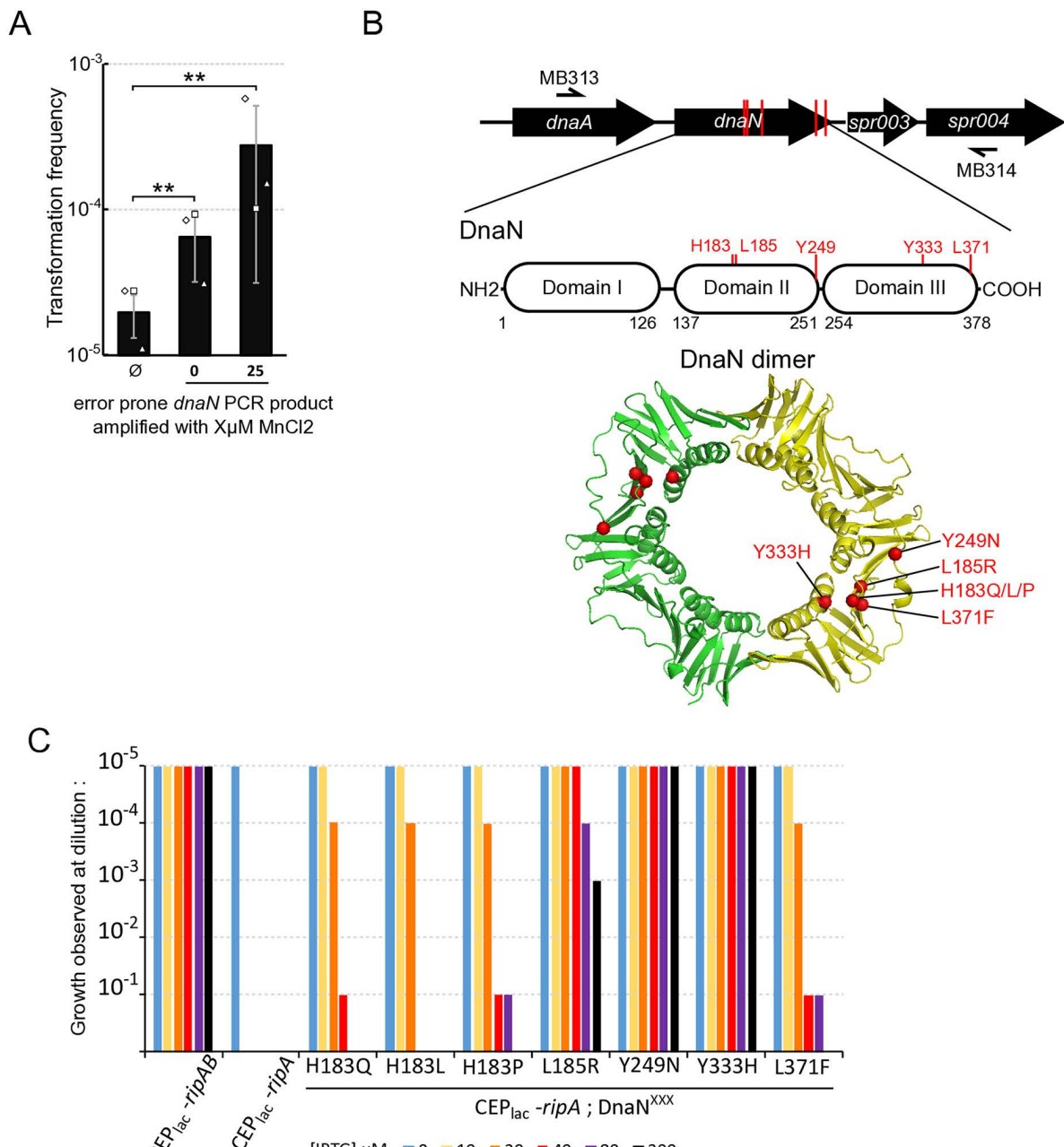

**Fig 4. Mutations in DnaN confer different levels of resistance to RipA toxicity.** A. Strain R5086, expressing *ripA* under the control of an IPTG inducible promotor was transformed with different error prone PCR fragments amplified from *dnaN locus* either without or with 25 μM of MgCl$_2$ to reduce polymerase fidelity. Transformant cells were plated in CAT agar supplemented with 4% horse blood containing 20μM IPTG. CFU were numerated after overnight incubation at 37 °C. Diamonds, squares, and triangles indicate the dispersion of the experimental results. B. Top panel. Schematic representation of the *dnaN locus* and its environment amplified by primers MB313 and MB314. Red vertical lines indicate the position of mutated residues that supress RipA toxicity. Positioning of the different suppressive mutations on the primary structure (middle panel) and on the tertiary 3D structure of DnaN, PDB ID code 2AWA, (bottom panel) PBD. C. Efficiency of plating of strains expressing *ripA* under the control of an IPTG inducible promotor in a *dnaN* wild type genetic background (R5139) or in a DnaN suppressor mutations genetic background (R5165 to R5171). The strain expressing *ripA-ripB* under the control of IPTG was used as a growth control (R5140). Cells were plated on CAT agar supplemented with 4% horse blood containing different concentrations of IPTG. For clarity, only a single spot test is presented as a histogram. Two independent determinations were made on different days are presented in S4 Fig.

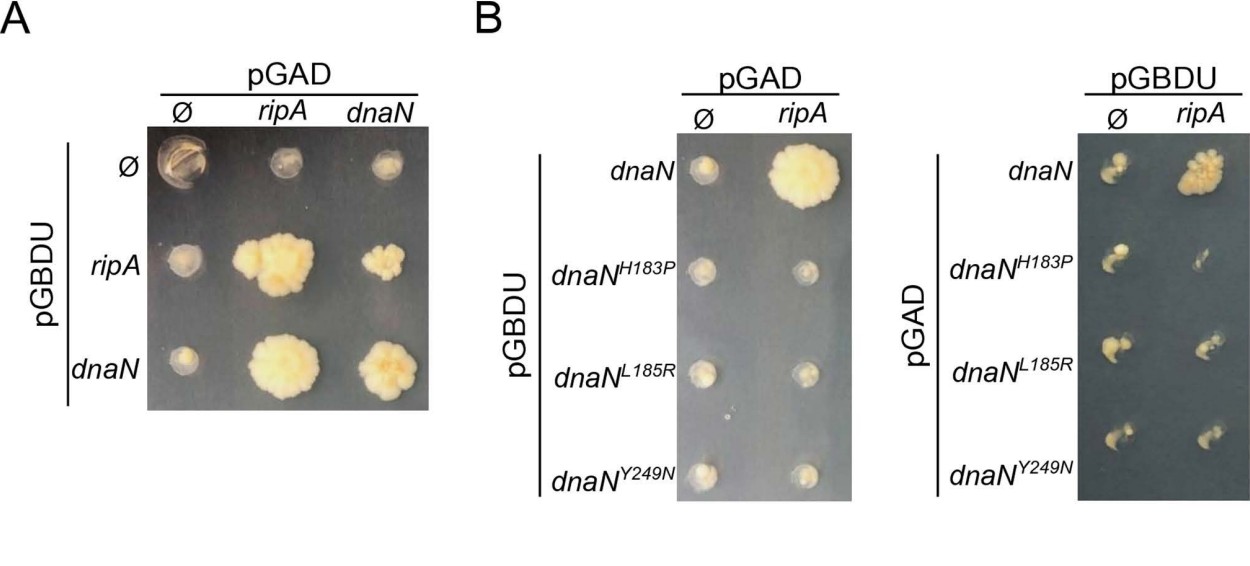

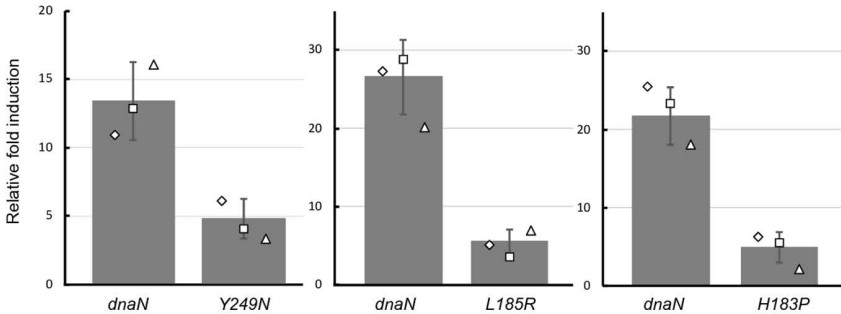

**Fig 5. Suppressor mutations in DnaN reduce interactions with RipA in Yeast-two-hybrid system and split-luciferase assay.** A. Yeast-two-hybrid matrices produced to test interactions between RipA and DnaN. B. Yeast-two-hybrid matrices performed to test interactions between RipA and DnaN suppressive alleles. For clarity, only a single assay is shown here; however, the three independent assays performed on different days are presented in S5 Fig. C. Split-luciferase assay comparing cellular proximity of RipA with DnaN (R5381), DnaN$^{H183P}$ (R5566), DnaN$^{L185R}$ (R5567), DnaN$^{Y249N}$ (R5568). Luminescence signal is expressed as a value relative to that obtained in strains R5380 and R5570, which each express only one of the potential interactors fused to the small luciferase subunit, while the large subunit is expressed alone. The two reference strains display similar signal levels.

As a negative control, we also used a strain with the same background, but unable to produce CSP (*comC0* strain). After the allotted time, these cells were mixed (1:1 ratio) with naive wild-type cells carrying the *comC*::*luc* reporter (strain R825). Almost instantaneously, wild-type cells switched to competence when mixed with RipA-producing capable of producing CSP, whereas no change was observed for wild-type cells alone or mixed with *comC0* cells (Fig 7D). These observations were complemented by measuring the ability of strain R825 to transform. Under these conditions, transformation can only occur in the presence of cells that produce *ripA* and are capable of producing CSP (Fig 7D, bottom panel). Altogether these results show that the RipA-producing cells, although rapidly losing their viability, remain metabolically active and produce sufficient CSP to propagate the competence to naive cells and promote their ability to transform.

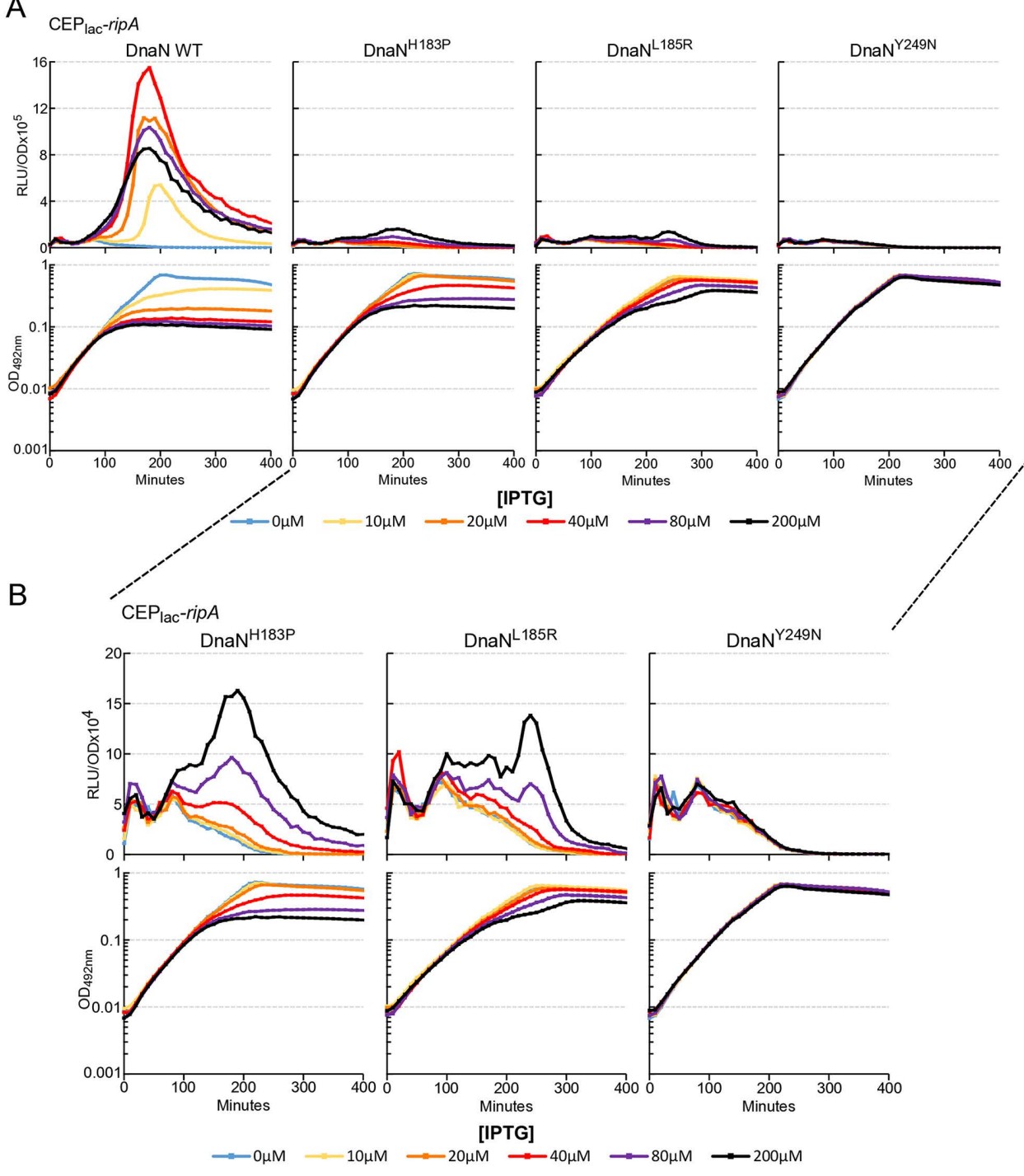

**Fig 6. Suppressor mutations in DnaN impair RipA dependant competence development.** A. *comCDE* expression was monitored in strains express-ing *ripA* under the control of an IPTG inducible promotor in a wild type *dnaN* genetic background (R5138) or in strains carrying suppressive alleles of *dnaN*, H183P (R5167), L185R (R5168), Y249N (R5169), conferring respectively weak, medium and strong suppressive phenotype. Cells were grown in C+Y medium at 37°C with increasing amounts of IPTG from the first measurement at time 0. Top panels: luciferase activity expressed in relative light units per OD (RLU/OD). Bottom panels: corresponding growth curves. For clarity, only a single assay is shown here; however, the three independent assays performed on different days are presented in S6 Fig. B. zoom in on transcription induction curves (A) in strains carrying suppressor mutations.

## Discussion

### ClpX and ClpP use independent pathways to repress competence

We found that the RipA toxin is responsible for the induction of competence generated by ClpX depletion but not ClpP depletion. Since deletion of *ripA* slightly improves the growth of a *clpP* strain, it is not possible to exclude the possibility of a partial link between RipA and ClpP, but ClpP-mediated induction of competence appears to be almost independent of RipA. It is very likely that ClpP, in association with other accessory proteins like ClpE and ClpC, ensures the homeostasis of several competence-regulating proteins, such as ComX or ComW respectively as previously demonstrated [36] or hypothesized, such as ComE [23]. Liu *et al.*, demonstrated that among the ClpP accessory proteins, only depletion of ClpX by CRISPRi was capable of inducing competence [24]. However, it should be kept in mind that the intensity of protein depletion generated by CRISPRi can be extremely variable from one gene to another or from one guide to another [37]. In addition, the quantity of accessory proteins required to ensure repression of competence can also be extremely variable depending on the protein. Understanding the role of ClpP in competence repression will require further investigation.

### How are RipA, RipB and ClpX related?

As *ripA* is responsible for *clpX* essentiality, it has already been hypothesised that RipA is a toxin that is impaired in its activity by ClpX [26]. We have furthermore shown here that the essential nature of RipB is also linked to the presence of RipA (Figs 2 and 3). The C-terminus of RipB protein shares homology with ImmA, a Zn metalloprotease involved in regulation of excision and transfer of the mobile genetic element ICEBs1 [38,39]. The N-terminus contains a transcriptional regulator domain of the XRE-family, sharing homology with the HipB antitoxin. As activity of RipA was impaired by trans-expression of *ripB*, it is reasonable to suggest that RipB acts as an antitoxin and that RipAB belongs to the large Rosmer TA family [40]. However, the RipAB system is not autonomous, since the action of RipB is strictly dependant on *clpX* integrity. This could be explained by different mechanisms. Firstly, we could assume that ClpX somehow modulates the toxin/antitoxin activation cycle. The involvement of ClpX in managing the function of a toxin-antitoxin system has already been reported, but until now its role has been described as activating the degradation of the antitoxin and therefore releasing the toxin [41–43]. In this work, ClpX plays a negative role in the action of the toxin. Given that RipB overexpression makes ClpX dispensable, it is tempting to propose that ClpX acts as a catalyst of RipB activity, either by enhancing RipB function or by presenting RipA to RipB (Fig 8). However, because toxin–antitoxin systems are often autoregulated, frequently through their antitoxin, we cannot exclude the possibility that RipB overexpression reduces *ripA* transcription, thereby limiting toxicity. Other more indirect relationships might also be involved, for example, in a ClpX-deficient genetic context, DnaN could to be more sensitive to the action of RipA. To distinguish between these hypotheses, dedicated experiments will be required in the future.

### The DnaN sliding clamp is a target of RipA

Given that RipA toxicity can be counteracted by mutations in the DnaN gene and that the same mutations abolish the Y2H interaction and split-luciferase between RipA and DnaN, it is reasonable to propose that the sliding clamp is the target of RipA. The most obvious molecular action would be that RipA affects the main action of the sliding clamp, by disrupting DNA replication. Indeed, the yeast two-hybrid and split-luciferase interaction assays, together with the disappearance or mislocalization of YFP–DnaX foci, support the notion of a potential perturbation of DNA replication. Furthermore, phenotypes associated with overexpression of genes located near the origin of replication are comparable to those observed upon treatment with a DNA polymerase inhibitor such as HPUra [19]. Replicative stresses are clearly known to be factors that induce competence [16,17,19,44], which is in line with our results showing that RipA production induces competence and that suppressive mutations in *dnaN* strongly antagonise it. However, we cannot rule out the hypothesis that DnaN is

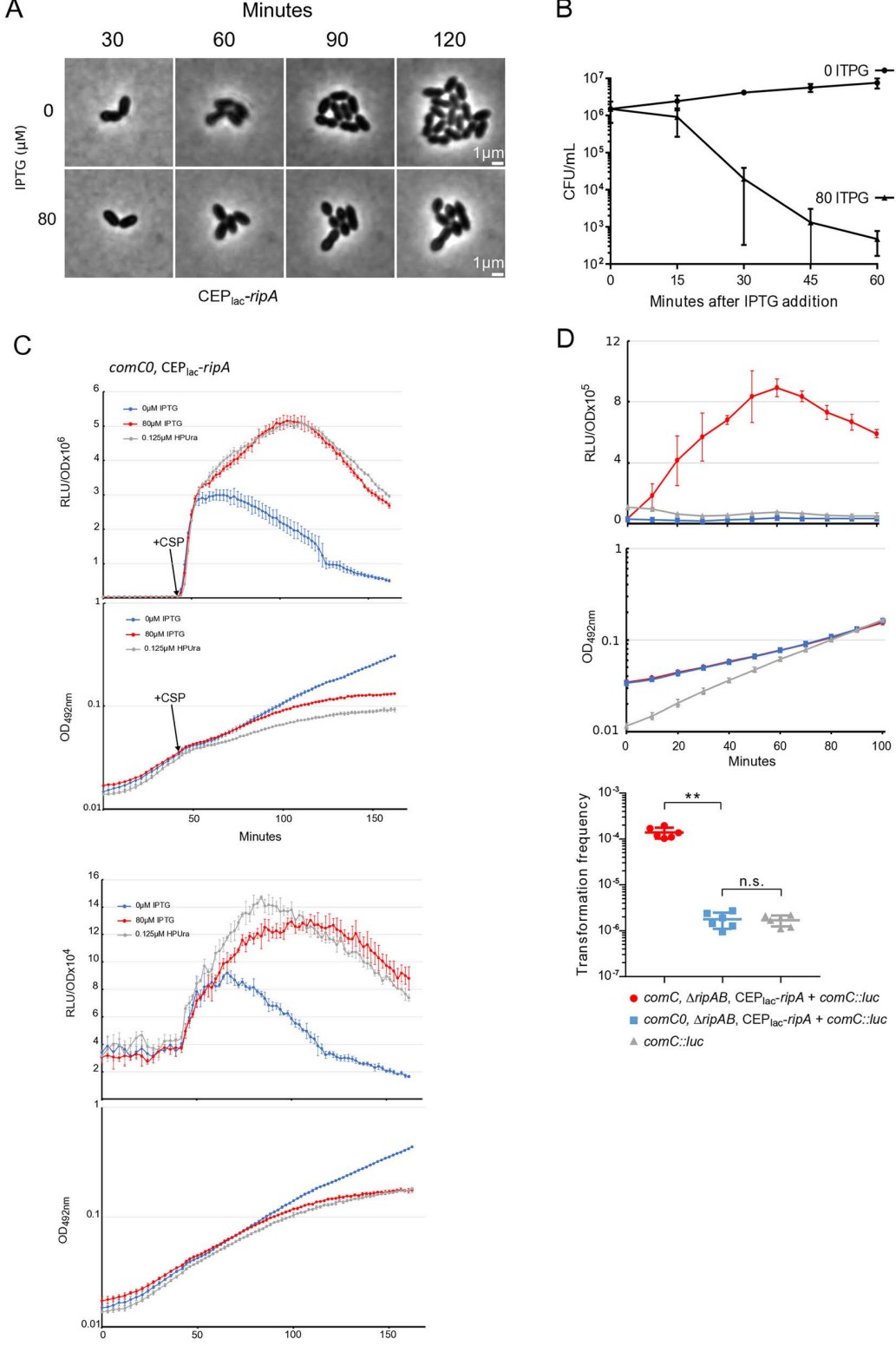

**Fig 7. RipA toxicity allows propagation of competence to naive cells.** A. Phase contrast time-lapses of the strain expressing *ripA* under the control of an IPTG inducible promotor (R5139) without or with 80µM of IPTG (respectively top and bottom panels). B. Ability of R5204 strain to generate colony

after IPTG exposure. R5204 strain was exposed or not to 80µM of IPTG for different times (X-axis), washed with fresh medium and plated on Agar medium without IPTG. C. Responsiveness of R5089 strain to CSP after one hour exposition or not to IPTG. The two upper panels show the transcriptional activity of *comC*::*luc* and the associated growth curve in the presence of CSP. The two lower panels show the equivalent measurements in the absence of CSP. At T0, cells were exposed to IPTG (red lines), HPUra (grey lines). Standard deviations represent the variation observed across three biological replicates. D. Mixed culture of strains expressing *ripA* under the control of an IPTG inducible promotor, R5204 (*comC*+) or R5086 (*comC₀*), with R825 strain used as competence reporter cells through its *comC*::*luc, comC* construct. Top panels: luciferase activity of R825 strain alone (grey) or mixed with R5204 (red) or R5086 (blue). Middle panels: corresponding growth curves. For clarity, only a single data set, representative of at least three independent determinations made on different days, is presented. Bottom panel. R825 transformation frequency that occurs during mixed cultures. PCR DNA conferring streptomycin resistance was added at T0. The cells were then collected at T100 and plated into agar medium containing erythromycin and streptomycin.

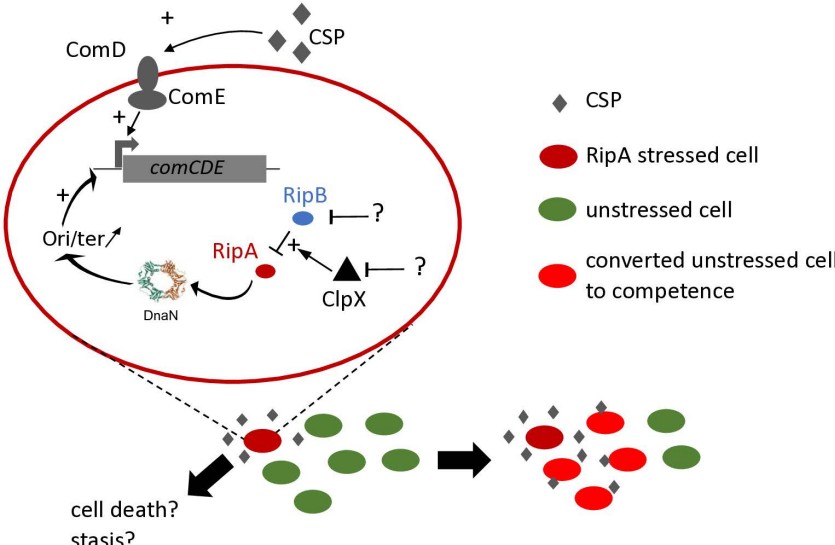

**Fig 8. Working model of RipA action on competence development at single cell and population level in *S. pneumoniae*.**

an intermediate target of RipA and that its final target would be a partner of the sliding clamp. DnaN is known to act as a hub for interactions between different proteins such as MutL and MutS, DNA polymerases or ligases [45]. The majority of these interactions occur via a hydrophobic pocket localized on the Pol face of the Clamp [32]. All the suppressive mutations uncovered in this work are localized in the hydrophobic pocket close to the interaction zone. It is therefore conceivable that RipA disrupts the dynamics of the sliding clamp interactome, thereby inducing perturbations in the replicative machinery.

## Biological role of RipA in *S. pneumoniae*

Several TA systems affecting replication have been discovered mostly affecting topoisomerases [46,47], but very few directly affect the sliding-clamp [48]. For example, SocB, the toxin of the SocAB type VI TA, was the first toxin described to target the sliding-clamp in *Caulobacter crescentus* [49]. The discovery of its target was also achieved using genetic approaches. The mutations suppressing the action of SocB were located in the same hydrophobic pocket of the sliding clamp as the one supressing RipA toxicity. In addition, it is interesting to note that SocB induces the SOS response system by altering replication in *C. crescentus*; while in *S. pneumoniae*, a bacterium lacking an SOS response system [49], *ripA* expression induces the development of competence, a physiological state frequently proposed as a replacement for

the SOS system [19,44,50,51]. However, consultation of alignment tools [52] did not reveal any sequence or structural homology between SocB and RipA, nor between their respective antitoxins. Taken together, these elements suggest a functional convergence of these TA systems towards a central target of the cell cycle.

However, one notable difference needs to be highlighted: while the induction of the SOS system remains an individual behaviour, the induction of competence leads to a population behaviour. In fact, it is well established that certain replicative stresses induce competence in *S. pneumoniae* and in this way, a fraction of the population subjected to RipA stress could transmit this signal through the production of CSP to the whole of the unstressed population, through a potential sacrificial behaviour (Fig 8). In the end, the switch to competence allows the generation of a heterogeneous population, both physiologically and genetically, allowing the emergence of potentially better adapted individuals [16]. The fate of the inducer cells is of particular interest, as it will provide clues as to the biological signals that activate RipA. Indeed, the transition of these cells to a stasis state could be consistent with the ability of antibiotics to induce competence [17]. In such a model, stressed cells that activate RipA trigger two mechanisms that allow them to transition to a potentially persistent state while enabling other cells to become competent. This could be relevant, given that competence enables the population to tolerate certain antibiotics more effectively [16]. Alternatively, sacrificial behaviour could limit the spread of danger while warning neighbouring cells of potential stress. This behaviour is reminiscent of phage resistance mechanisms, in which TA systems are involved [48,53]. This suggests that we should explore the possibility of a previously unknown link between competence and phages in *S. pneumoniae*.

## Materials and methods

### Strains and growth media

All *S. pneumoniae* strains used were derived from R800 strain [54] and are listed in S1 Table.

Standard procedures for transformation and growth media were used [55]. Briefly, pre-competent cells were treated at 37°C for 10 min with synthetic CSP1 (100 ng mL$^{-1}$) to induce competence, then exposed to transforming DNA for 20 min at 30°C. Transformants were then plated in CAT agar supplemented with 4% horse blood and incubated for 120 min at 37°C. Transformants were then selected by addition of a second layer of agar medium containing the appropriate antibiotic and incubated overnight at 37°C. Antibiotic concentrations (µg mL$^{-1}$) used for selection were: chloramphenicol (Cm), 4.5; kanamycin (Kan), 250; spectinomycin (Spc), 100; streptomycin (Sm), 200, gentamycin (G), 40 and erythromycin (E), 0.1. Unless otherwise described, pre-competent cultures were prepared by growing cells to an OD$_{550nm}$ of 0.1 in C+Y medium (pH 6.8). Then cells were 10-fold concentrated in C+Y medium supplemented with 15% glycerol and stored at –80°C.

### Deletion and invalidation mutagenesis

Deletion or invalidation mutagenesis was based on strand overlap extension (SOE) [56]. Briefly, primers MP170 and MP173 were used to generate PCR fragments carrying spectinomycin or kanamycin resistance gene from plasmids pr412 (SpcR) or pr413 (KanR) previously described [57]. The two PCR fragments that flank the integration site of the resistance gene were amplified with the specific primer pairs described in S2 Table. These pairs are composed of a primer defining the integration site and carrying the sequence complementary to MP170 or MP173 in its 5' region and a distal primer at around 1–2 kb. The three PCR amplified fragments were purified and used as template to produce a unique PCR fragment using the two distal primers. The resulting fragment was used to transform a recipient strain as described above. The transformed strains were selected by appropriate antibiotic selection and the transformed locus was sequenced (Eurofins genomics).

### IPTG-dependent expression platform construction

Two gene-expression platforms were routinely used [58]. Both allow the expression of a gene of interest under the control of an IPTG-inducible promoter. Upstream, the lacI gene is positioned in the opposite transcriptional orientation, while

downstream lies an antibiotic-resistance gene. These platforms are inserted either at the *ami* locus, where they are linked to kanamycin resistance and called CEP$_{lac}$, or at the cpsN locus, where they are associated with erythromycin resistance and Called CEPII$_{lac}$. SOE was used to generate expression platforms CEP$_{lac}$-*ripA*, CEPlac-*ripB* or CEPlac-*ripAB*. Primers pairs MB293-MB294 and MB295-MB291 were used to amplify an expression platform from strain R3833 previously described [28]. Internal PCR fragments carrying genes of interest were amplified from R800 strain using the following primers pairs MB296-MB297 (*ripAB*), MB296-MB298 (*ripA*), and MB299-MB297 (*ripB*). Each of these fragments was used with MB293-MB294 and MB295-MB291 primer pairs as templates to generate a unique PCR fragment with primers MB293 and MB291. This final fragment was used to transform strains of interest as described above (kanamycin selection).

The CEPII$_{lac}$::*ripA* platform (erythromycin) was generated by transferring the CEP$_{lac}$::*ripA* platform to the *cpsN locus* but without downstream kanamycin resistance. Briefly, MM56 and MM57 were used to amplify the upstream part of the platform from strain R5139. Flanking *cpsN locus* and erythromycin gene resistance were amplified using YA09-MM58 et YA14-MM59 primer pairs with R4631 strain as a template. This strain contains CEPII$_{lac}$-*dprA*-mturquoise (ery$^{R}$) [34]. These three PCR fragments were used as templates to amplify a single PCR fragment using primers YA09-YA14. This fragment was used to transform strain R5198 (erythromycin selection).

### CRISPRi depletion

Strains carrying the CRISPRi system were constructed as described previously [24]. Plasmids carrying *lacI* repressor (pPEPY-PF6-*lacI*), Cas9 enzyme (pJWV102-PL-*dCas9*) and expressing interfering sgRNA (pPEPX-P3-sgRNAluc) were purchased from addgene (respectively #85589, #85588 and #85590). Plasmids pPEPX-P3-sgRNA-*clpX* and pPEPX-P3-*clpP* were generated using the In-Fusion HD cloning kit (Takara), as described previously [24]. Primers OIM134 and OIM135 were used to amplify the pPEPX-P3-sgRNA backbone from the pPEPX-P3-sgRNAluc template. Guide RNA sequences were produced using primer pairs MB551–MB552 (*clpX*) and MM54–MM55 (*clpP*), as reported earlier [24]. The resulting plasmids were sequentially integrated into the genome of strain R825 through successive transformation and homologous recombination. These strain was then transformed sequentially by pPEPY-PF6-*lacI* (gentamycin resistance, at *prsA* locus) then pJWV102-PL-*dCas9* (tetracycline resistance at *bgaA* locus) and pPEPX-P3-sgRNA-*clpX* or pPEPX-P3-*clpP* (kanamycin resistance at *ami* locus).

### Monitoring of growth and luciferase expression

To monitor *comCDE* operon expression, we used a previously established transcriptional fusion [57]. Briefly, a fragment of the *S. pneumoniae comCDE* promoter (HindIII–BamHI) was inserted upstream of the *luc* gene in a non-replicative plasmid carrying erythromycin resistance. Homology-dependent integration of this recombinant plasmid into the pneumococcal chromosome was then selected using erythromycin.

For the monitoring of growth and luciferase expression, precultures were gently thawed and aliquots were inoculated (unless otherwise described) at OD$_{550nm}$ of 0.005 in luciferin-containing C+Y medium (pH 7) as previously described [55] and distributed into a 96-wells (300 μl per well) white microplate with clear bottom (Corning). Relative luminescence units (RLU) and OD$_{492nm}$ values were recorded at defined time points throughout incubation at 37°C in a Varioskan luminometer (ThermoFisher).

### DnaN error prone PCR and screening

DnaN *locus* was amplified from R800 genomic DNA using primers MB313 and MB314 and DreamTaq DNA Polymerase (thermoFisher scientific) as recommended by the supplier with or without adding 25μM of MnCl$_2$ to reduce the fidelity of the enzyme. The resulting fragments were used to transform strain R5086 as described above. After an incubation of

120min in CAT medium at 37°C, transformant cells were plated and grown 24 hours on CAT agar supplemented with 4% horse blood and selected with 20μM IPTG. DnaN *locus* was amplified on potential suppressor clones using primers MB313 and MB314 and a high-fidelity DNA polymerase, Primestar max DNA polymerase (Takara). PCR fragments were used to transform R5086 as described above (selection with 20μM IPTG) to confirm that suppressive mutations were linked to DnaN *locus*. Relevant PCR fragments were sequenced (Eurofins genomics).

### 3D mapping of suppressive mutations on DnaN

Suppressive mutations were positioned on 3D DnaN structure from Protein Data Bank (PDB), www.pdb.org (PDB ID code 2AWA) using Pymol software V0.99.

### Yeast two hybrid

Gene-coding sequences of *S. pneumoniae* Spr1630 and DnaN proteins were PCR amplified using R1501 DNA as template, and inserted in Gal4-based plasmids, PGAD-C1 and PGBD-C1 [59], using the In-Fusion HD cloning kit (Takara). Primers used were MM30 and MM31 (RipA), MM28 and MM29 (DnaN) and OCN424 and OCN425 (PGAD-C1 and PGBD-C1).

Resulting plasmids were then transformed independently in the yeast strains PJ69-4a and PJ69–4α. *Saccharomyces cerevisiae* cells expressing *S. pneumoniae* proteins as GAL4 Binding Domain (BD) fusions were mated with cells expressing some of these proteins as GAL4 Activating Domain (AD) fusions. Binary interactions were identified by growth of diploid cells after 8 or 20 days at 30°C on synthetic complete medium [60] lacking leucine, uracil and histidine (to select expression of the HIS3 interaction reporter). Controls with empty vector plasmids (i.e., carrying only the BD or AD domain) were systematically included.

### Split luciferase assay

**Construction of the CEP$_{lac}$-*lgbit–ripA* platform.** Primers MB583 and MB584 were used on strain R4856 to PCR-amplify the region encoding the Nanoluciferase large subunit. These primers correspond to MB585 and MB586, which flank the intended insertion site. PCRs using primer pairs MB586-MB293 and MB585-MB291 were performed in strain R5086. The three resulting fragments were purified and assembled to generate a single fragment containing the Nanoluciferase large subunit. Transformation was carried out in strain R5085, followed by kanamycin selection to generate R5352.

**Construction of the CEPII$_{lac}$-*smBit-dnaN* platform.** The small subunit was fused to the N-terminus of DnaN through PCR amplification performed on the DnaN locus (strain R800) using primers MB597 and MB598. Amplification of the flanking fragments encoding the CEPII$_{lac}$ expression platform was performed in strain R5239 using the primers pairs MB594–YA09 and R599–YA14. The three fragments were then amplified together using primers YA09 and YA14 and transformed into strain R5352 or R5085, with selection based on erythromycin resistance to generate R5381 or R5370.

**Construction of the CEP$_{lac}$-*lgbit* platform.** The CEP$_{lac}$ platform producing the large Nanoluciferase subunit alone was constructed as follows. Primers pairs MB595-MB291 andMB596-MB293 were used to amplify the large subunit without *ripA* from strain R5086. The two fragments were then amplified and fused by PCR using oligonucleotides MB293 and MB291. Strain R5370 was transformed with this PCR product and selected for kanamycin resistance to gene to generate strain R5380.

Split luciferase assays were carried out as previously described [61–63], with modifications. Briefly, pneumococcal cells were grown in a C + Y medium to an $OD_{550nm}$ of 0.05. IPTG was added to a final concentration of 200 μM, and cultures were incubated for 1h. Then, 80 μL of cells were mixed with 20 μL of 1% NanoGlo substrate (Promega). Luminescence was measured 30 times at 30-s intervals using a plate reader (VarioSkan luminometer, Thermo Fisher). Luminescence values obtained after 200 seconds of measurement were summed and expressed relative to those obtained with the negative controls R5380 and R5570.

## Fluorescence microscopy and image analysis

To visualize cells by epifluorescence microscopy, pneumococcal precultures were grown in C+Y medium (without glucose and saccharose) supplemented with 0.03% maltose at 37 °C to an $OD_{550}$ of 0.07. The culture was then split in two, and IPTG (200 µM) was added to one half. The cells were incubated at 37 °C for 30, 90, 120, or 180 minutes. After incubation, 2 µL samples were spotted onto a microscope slide containing a slab of 1.2% C+Y agarose, as previously described [64] before imaging. Phase contrast and fluorescence microscopy were performed as previously described. Images were processed using the Nis-Elements AR software (Nikon) [65]. Cells were segmented using Omnipose [66]. Images were analysed using MicrobeJ, a plug-in of ImageJ [67].

## Mixed culture assay

Strains R5204 and R5086 were inoculated at 0.04 $OD_{550nm}$ in C+Y medium and grown in the presence of 80 µM of IPTG for 120 minutes. In parallel R825 strain was inoculated at 0.04 $OD_{550nm}$ in C+Y medium (pH 7) and grown for 60 minutes. Cells R5204 or R5086 were mixed with R825 (volume to volume) to an expected ratio of 1–1. R825 luciferase activity was then monitored. PCR amplified DNA carrying *rpsL41* allele (Streptomycin resistance) was added at the time of mixing. 100 minutes after mixing a culture sample was collected and plated on agar medium supplemented with erythromycin and streptomycin to determine the percentage of transformation of strain R825.

## Statistical tests

Pairwise comparisons were done with a nonparametric MannWhitney test. P values were displayed as follows: ***, $0.0001 < P < 0.001$; **, $0.001 < P < 0.01$; *, $0.01 < P < 0.05$; ns, $P > 0.05$.

## Supporting information

**S1 Fig. Depletion of *clpX* or *clpP* induces competence.** A. *comCDE* expression was monitored in strains carrying CRISPRi guide RNA targeting *clpX*, R4993 (left) or *clpP* R5203 (right). Cells were grown in C+Y medium at 37°C with or without IPTG, as shown in the colour key. Culture was initiated 100 minutes prior to the first measurement at time 0. *comCDE* expression values are expressed in relative light units per OD (RLU/OD) in the top panel and the corresponding growth curves are reported as $OD_{492nm}$ in the bottom panel. Data represented as Mean±standard deviation of triplicate repeats. B. Identical to A in Δ*ripA* strains carrying CRISPRi guide RNA targeting respectively *clpX*, R4995 (left) or *clpP*, R5250 (right). Data represented as Mean±standard deviation of triplicate repeats. C. Comparison of the growth curves of ClpP-depleted strains in the presence or absence of the *ripA* gene, plotted on a linear y-axis. Red: Δ*ripA*; Black: *ripA* wild-type. (TIF)

**S2 Fig. *ripA* expression induces competence development.** A. Replicates of Fig 2B. *comCDE* expression was monitored in strains expressing *ripA* (R5139), *ripB* (R5138) or *ripA-ripB* (R5140) under the control of an IPTG inducible promotor. Cells were grown in C+Y medium at 37°C with increasing concentration of IPTG from the first measurement at time 0. Top panels: luciferase activity expressed in relative light units per OD (RLU/OD). Bottom panels: corresponding growth curves. B. Deletion of the *ripA* or *ripAB* locus reduces competence induction efficiency. R825, R4422, and R4423 strains were cultured in C+Y medium under conditions near the competence induction threshold (mild medium acidification). For each experiment, four independent cultures per strain were monitored, and the proportion of cultures undergoing competence induction was quantified. The experiment was performed three times on separate days. The standard deviation represents the variability in competence induction frequencies across the three independent experiments. (TIF)

**S3 Fig. ClpX is genetically linked to RipAB, a toxin–antitoxin system involved in competence development.** A. Induction of basal transcription levels of *comA and tRNA-arg5. comC0*-derived strains producing RipA under an IPTG-inducible promoter were grown in the presence of increasing IPTG concentrations. *comA* (R5572) and *tRNA-arg5* (R5571) transcription levels were monitored through luciferase activity (left panels). For clarity, only one representative experiment is shown; however, the areas under the curves were quantified from three independent experiments and are presented as bar graphs with their respective standard deviations as shown in the middle panels. Right panels: corresponding growth curves. B. Overexpression of RipB compensates for the loss of *clpX*. Strain R5565 overexpressing *ripB* under the control of an IPTG-inducible promoter was transformed with a *clpX* deletion PCR fragment and then plated in the presence or absence of 200 μM IPTG. R4796 strain, deleted for *ripA*, was used as a positive control for transformation (n = 3).
(TIF)

**S4 Fig. Mutations in DnaN confer different levels of resistance to RipA toxicity.** Efficiency of plating of strains expressing *ripA* under the control of an IPTG inducible promotor in a *dnaN* wild type genetic background (R5139) or in a DnaN suppressor mutations genetic background (R5165 to R5171). The strain expressing *ripA-ripB* under the control of IPTG was used as a growth control (R5140*)*. Cells were plated on CAT agar supplemented with 4% horse blood containing different concentrations of IPTG. For clarity, each single spot test is presented as a histogram.
(TIF)

**S5 Fig. Yeast-two-hybrid matrices performed to test interactions between RipA and DnaN suppressive alleles.** Three independent yeast two-hybrid experiments were performed in triplicate, testing the interaction in both possible orientations.
(TIF)

**S6 Fig. Suppressor mutations in DnaN impair RipA dependant competence development.** Two replicates of experiments displayed Fig 6*. comCDE* expression was monitored in strains expressing *ripA* under the control of an IPTG inducible promotor in a wild type *dnaN* genetic background (R5138) or in strains carrying suppressive alleles of *dnaN*, H183P (R5167), L185R (R5168), Y249N (R5169), conferring respectively weak, medium and strong suppressive phenotype. Cells were grown in C+Y medium at 37°C with increasing amounts of IPTG from the first measurement at time 0. Top panels: luciferase activity expressed in relative light units per OD (RLU/OD). Bottom panels: corresponding growth curves.
(TIF)

**S7 Fig. Impact of RipA on cell cycle and replication fork.** A. Phase contrast time-lapses of the strain expressing *ripA* under the control of an IPTG inducible promotor (R5139) without or with 80μM of IPTG (respectively top and bottom panels). Two independent experiments. B. Microscopy-based cell enumeration showed in A. Means of two independent experiments are shown, together with their standard deviations. The initial cell number at T0 was normalized to 100. Actual initial cell counts were: without IPTG, n = 398 and 372; with IPTG, n = 364 and n = 475. C. Representative microscopy images of the R5260, expressing YFP-DnaX and carrying Plac-*ripA*, in the presence or absence of IPTG. Blue arrows; cells without focus, green arrows; cells with polar focus. D. Quantification of the number of YFP-DnaX fluorescent foci per cell in the R5260 grown with or without IPTG. Number of cells analysed at T0: n = 710, T30: n = 720 + 962, T90: n = 518 + 1675, T120: n = 2382 + 819, T180; n = 1312 + 1130. With IPTG, T30: n = 766 + 1390, T90: n = 2542 + 1442, T120: n = 3119 + 1266, T180: n = 3843 + 1889. Results without IPTG; 0 focus T30: 12.26 ± 3.98, T90: 6.51 ± 2.53, T120: 10.08 ± 5.77, T180: 9.34 ± 2.32. 1 focus T30: 81.84 ± 0.34, T90: 84.34 ± 2.98, T120: 80.41 ± 0.10, T190: 77.46 ± 1.91. 2 foci: T30: 5.78 ± 3.61, T90: 8.62 ± 0.37, T120: 8.88 ± 4.78, T180: 12.62 ± 0.17. Results with IPTG; 0 focus: T30: 11.13 ± 0.23, T90: 15.64 ± 0.9, T120: 24.22 ± 0.15, T180: 27.36 ± 6.08. 1 focus: T30: 82.79 ± 3.17, T90: 75.27 ± 2.31, T120: 68.46 ± 1.31, T180: 66.09 ± 4.31. 2 foci: T30: 5.68 ± 3.05, T90: 8.83 ± 0.86, T120: 7.21 ± 1.31, T180: 6.31 ± 1.67. E. Localization of YFP-DnaX fluorescent foci in R5260 under IPTG-induced and non-induced conditions. Number of foci localized at T0: n = 710,

T30: n = 714 + 850, T90: n = 527 + 1752, T120: n = 2592 + 749, T180; n = 1402 + 1154. With IPTG, T30: n = 752 + 1288, T90: n = 2314 + 1390, T120: n = 2592 + 1061, T180: n = 3270 + 1394. Results without IPTG; midcell: T30: 80.25 ± 0.19, T90: 75.12 ± 2.18, T120: 75.06 ± 0.79, T180: 77.28 ± 2.07. Polar: T30: 13.17 ± 1.00, T90: 12.61 ± 3.07, T120: 12.40 ± 1.67, T180: 14.20 ± 2.83. Betwixt: T30: 6.57 ± 0.81, T90: 12.27 ± 0.89, T120: 12.54 ± 2.46, T180: 8.52 ± 0.77. Results with IPTG: midcell: T30: 79.52 ± 2.06, T90: 67.07 ± 1.40, T120: 55.68 ± 2.03, T180: 57.10 ± 2.93. Polar: T30: 14.37 ± 0.2, T90: 28.24 ± 0.77, T120: 37.62 ± 2.46, T180: 35.57 ± 3.75. Betwixt: T30: 6.03 ± 2.38, T90: 4.69 ± 0.63, T120: 6.67 ± 0.43, T180: 7.32 ± 0.81. F. Comparison of the kinetics of emergence of polar foci and of cells lacking foci in the R strain under IPTG induction. Standard deviations represent the variation observed across two biological replicates.
(TIF)

**S1 Table. Strains used in this study.**
(DOCX)

**S2 Table. Oligonucleotides used in this study.**
(DOCX)

## Acknowledgments

We would like to extend our special thanks to Calum Johnston and Hélène Cordier for their critical review of the manuscript.

## Author contributions

**Conceptualization:** Mickaël Maziero, Patrice Polard, Mathieu Bergé.

**Formal analysis:** Mickaël Maziero, Dimitri Juillot, Isabelle Mortier-Barrière, Rut carballido-lopez, Patrice Polard, Mathieu Bergé.

**Investigation:** Mickaël Maziero, Dimitri Juillot, Isabelle Mortier-Barrière, Mathieu Bergé.

**Supervision:** Rut carballido-lopez, Patrice Polard, Mathieu Bergé.

**Validation:** Nathalie Campo.

**Writing – original draft:** Mathieu Bergé.

**Writing – review & editing:** Mickaël Maziero, Dimitri Juillot, Rut carballido-lopez, Nathalie Campo, Pierre Genevaux, Patricia Bordes, Patrice Polard, Mathieu Bergé.

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
