## [Decision Letter · Decision Letter 0]

25 Sep 2025

PGENETICS-D-25-00966

A toxin/antitoxin system targeting the replication sliding-clamp induces competence in Streptococcus pneumoniae.

PLOS Genetics

Dear Dr. BERGE,

Thank you for submitting your manuscript to PLOS Genetics. After careful consideration and based on the reviewers comments, we believe the work has merit but that additional work that provides more mechanistic insight would be necessary for publication. Specifically, this relates to demonstrating how RipA affects DnaN (reviewer 1 & 2), and the link between RipAB and ClpX (reviewer 1 & 2). Also make sure to clarify experimental replicates and statistical analyses. We invite you to submit a revised version of the manuscript that addresses these points and the other points raised during the review process.

Please submit your revised manuscript within 60 days Nov 24 2025 11:59PM. If you will need more time than this to complete your revisions, please reply to this message or contact the journal office at plosgenetics@plos.org. Please include the following items when submitting your revised manuscript:

We look forward to receiving your revised manuscript.

Kind regards,

Morten Kjos

Academic Editor

PLOS Genetics

Sean Crosson

Section Editor

PLOS Genetics

Aimée Dudley

Editor-in-Chief

PLOS Genetics

Anne Goriely

Editor-in-Chief

PLOS Genetics

**Journal Requirements:**

We have noticed that you have uploaded Supporting Information files, but you have not included a list of legends. Please add a full list of legends for your Supporting Information files after the references list.

**Reviewers' comments:**

Reviewer's Responses to Questions

**Comments to the Authors:**

Reviewer #1: This manuscript reports that ClpX, the ATPase subunit of the AAA+ protease, relies on the toxin RipA-which targets the sliding clamp DnaN and consequently disrupts bacterial DNA replication-to induce pneumococcal competence. This conclusion is based on appropriate molecular analyses of genetic mutagenesis, reliable competence assays, and protein-protein interaction studies. Overall, these findings provide new evidence emphasizing that DNA replicative stress serves as a signal to activate competence in S. pneumoniae. Although this is a carefully conducted study, several questions remain to be addressed.

1. The conclusion that the toxin RipA induces competence activation and growth arrest is based on its targeting of DnaN (the sliding clamp), thereby disrupting DNA replication. While this interpretation is reasonable, no direct evidence demonstrates actual inhibition of bacterial DNA replication. This point becomes more doubtful given the absence of any sequence or structural homology between RipA and SocB, the first toxin reported to target the sliding clamp in Caulobacter crescentus (line 285), nor between their respective antitoxins (line 293).

2. The authors demonstrated that RipA-triggered DNA replication arrest induces bacteria to become competent. This raises an interesting question: does the activation of competence alleviate the toxic effects of RipA itself?

3. To further demonstrated RipAB belongs to a TA system, it’s better to provide evidence of co-transcription of these two genes.

4. Line 259: Regarding the relationship between ClpX and RipAB, is it possible that the basal expression level of RipB is insufficient to counteract the toxicity of RipA? If so, overexpressing RipB might eliminate the dependence on RipA for ClpX function.

5. Fig. 4B: is there any clear annotation of the three domains of DnaN, which may indicate their functions?

6. Is there any clue that reveals the interaction mechanism between RipA and DnaN by predicted structural analysis, particularly RipA with the five different amino acids of DnaN (line 164).

7. The authors claim that bacterial multiplication was affected after 90 minutes of IPTG exposure, a conclusion based solely on the microscopy data presented in Fig. 7A. Statistical analysis of multiple microscopy fields through quantification of bacterial cell counts over the course of IPTG exposure would be necessary to support this assertion.

8. Lines 219 and Fig. 7C: Why did cells producing RipA with IPTG treatment for more than 60 minutes exhibit more rapid and higher competence induction in response to synthetic CSP? The induction of RipA by IPTG appears to enable at least a subset of pneumococcal cells to become competent. However, it is well-established that bacterial cells enter a recovery phase following competence, which seems inconsistent with the authors’ observation.

9. Line 259: ‘dependant’ should be ‘dependent’.

10. Line 288: ‘supressing’ should be ‘suppressing’.

Reviewer #2: my review is in the attached file

Reviewer #3: MS # PGenetics-D-25-00966, by Maziero et al. brings to light a new toxin/antitoxin system in S. pneumoniae that links the toxin component to competence for genetic transformation and stress responses. The tandem gene pair, which was noted as affecting the essentiality of ClpX nearly two decades ago, but with unknown functions, is now designated as RipA,RipB (for replication interfering protein). Thorough genetic and biochemical dissection in this manuscript now reveals, surprisingly, that the molecular target of the toxin component is DnaN, the sliding clamp of DNA replication.

A model of expository clarity, the submitted manuscript needs little further revision. However, careful review, perhaps best done by a first-time reader, would be rewarded by unearthing a few remaining typos.

A few examples:

l. 210. “exposure” is presumably intended, not “exposition”.

l. 317. The standard usage would be “strain R800” not “R800 strain,” as is used regularly throughout the text. Similarly, see lines 405, 609, 651, 653, 654…

**Have all data underlying the figures and results presented in the manuscript been provided?**

Reviewer #1: None

Reviewer #2: Yes

Reviewer #3: Yes

PLOS authors have the option to publish the peer review history of their article (what does this mean? ). If published, this will include your full peer review and any attached files.

**Do you want your identity to be public for this peer review?** For information about this choice, including consent withdrawal, please see our Privacy Policy .

Reviewer #1: No

Reviewer #2: No

Reviewer #3: **Yes: ** Donald Morrison

**Figure resubmission:**
---

## [Editor Report · Decision Letter 1]

14 Dec 2025

Dear Dr BERGE,

We are pleased to inform you that your manuscript entitled "A toxin/antitoxin system targeting the replication sliding-clamp induces competence in Streptococcus pneumoniae." has been editorially accepted for publication in PLOS Genetics. Congratulations!

Yours sincerely,

Morten Kjos

Academic Editor

PLOS Genetics

Sean Crosson

Section Editor

PLOS Genetics

Aimée Dudley

Editor-in-Chief

PLOS Genetics

Anne Goriely

Editor-in-Chief

PLOS Genetics

BlueSky: @plos.bsky.social

Comments from the reviewers (if applicable):

**Data Deposition**

http://datadryad.org/submit?journalID=pgenetics&manu=PGENETICS-D-25-00966R1

**Press Queries**

---

## [Editor Report · Acceptance letter]

PGENETICS-D-25-00966R1

A toxin/antitoxin system targeting the replication sliding-clamp induces competence in Streptococcus pneumoniae.

Dear Dr BERGE,

We are pleased to inform you that your manuscript entitled "A toxin/antitoxin system targeting the replication sliding-clamp induces competence in Streptococcus pneumoniae." has been formally accepted for publication in PLOS Genetics! Your manuscript is now with our production department and you will be notified of the publication date in due course.

With kind regards,

Anita Estes

PLOS Genetics

On behalf of:
